# Prediction of Typhoon Pathway Points Based on Zhang Extrapolation (ZE) Formula for Points 1 to 3

Chengwen Tang*, Litian Li†, Tao Wang*, Yunong Zhang*

*School of Computer Science and Engineering, Sun Yat-sen University, Guangzhou, China

tangchw3@mail2.sysu.edu.cn, {wangtao29, zhynong}@mail.sysu.edu.cn

†School of Artificial Intelligence, Sun Yat-sen University, Zhuhai, China

lilt26@mail2.sysu.edu.cn

*Abstract*—**This study evaluates the effectiveness of Zhang extrapolation (ZE) formulas, on the basis of equally spaced time point sequence, in predicting typhoon pathways using limited historical data. The ZE formulas for 1 through 3 points are tested on typhoons Saudel, Molave, and Krovanh, using a dataset from the Wenzhou typhoon network (WTN). Statistical analyses reveal that the point-1 extrapolation consistently offers superior accuracy, with mean absolute error and great-circle distance metrics within practical limitations. Comparisons with other methods show the proposed ZE formulas deliver higher precision, making them highly enhance applications in maritime safety and emergency response.**

*Index Terms*—**Typhoon prediction, Zhang extrapolation formulas, time sequence analyses, great-circle distance, mean absolute errors**

## I. INTRODUCTION

Typhoons, also known as tropical cyclones or hurricanes in different regions, are among the most destructive natural phenomena. They pose significant threats to life, property, and environment, particularly in coastal regions. Accurate prediction of typhoon pathways is crucial for mitigating these risks and enabling timely evacuations as well as preparations. Traditional typhoon prediction methods [1]–[3] rely on complex numerical models that require substantial computational resources and extensive historical data. However, these methods may not always be feasible in real-world scenarios where data is limited or unavailable.

Previous studies [4]–[9] have explored the use of Zhang extrapolation (ZE) formulas for time-sequence data to predict future values based on limited historical observations. These formulas, derived using Lagrange interpolation polynomials [4]–[9], offer a practical approach for predicting future data points in equally spaced time sequence. The ZE formulas have been shown to effectively approximate future values with minimal data, making them suitable for applications where data availability is constrained.

In this paper, we apply ZE formulas for points 1, 2, and 3 to the prediction of typhoon pathway points. The objective is to illustrate the practicality and real-world value of these formulas in predicting the trajectory of typhoons with limited historical data. We test ZE formulas on the pathways of three typhoons, namely typhoons Saudel, Molave, and Krovanh, using a dataset from the Wenzhou typhoon network (WTN). By comparing the extrapolation results with observed data,

we evaluate the accuracy and effectiveness of ZE formulas in predicting typhoon pathways.

The application of ZE formulas in this paper focuses on predicting future typhoon pathway points under extreme conditions, such as during maritime operations where external warning information may not be available. For instance, a ship encountering a typhoon at sea may need to estimate the next pathway point of the typhoon to avoid its impact, using only a small amount of recorded path data. This paper aims to provide a practical solution for such scenarios, enhancing maritime safety and emergency response.

The following sections describe the methodology, including the derivation of ZE formulas, the dataset used, and the statistical analyses of the extrapolation results. We compare the performance of ZE formulas with other prediction methods and provide a detailed discussion of their applicability and limitations. The results show that ZE formulas deliver higher precision and the effectiveness in predicting typhoon pathways, offering valuable insights for real-world applications.

TABLE I
ZE FORMULA USED IN TYPHOON PATHWAY POINT PREDICTION

| | | |
|---|---|---|
| **ZE Formula for Point 1** | **2 Terms** | $\vartheta_{k+1} = 2\vartheta_k - \vartheta_{k-1}$ |
| | **3 Terms** | $\vartheta_{k+1} = 3\vartheta_k - 3\vartheta_{k-1} + \vartheta_{k-2}$ |
| **ZE Formula for Point 2** | **2 Terms** | $\vartheta_{k+2} = 3\vartheta_k - 2\vartheta_{k-1}$ |
| **ZE Formula for Point 3** | **2 Terms** | $\vartheta_{k+3} = 4\vartheta_k - 3\vartheta_{k-1}$ |

To better fit the problem context, namely, predicting a future value with fewer historical data points, the number of terms in the selected ZE formulas (points 1, 2, and 3) is 2 or 3, as shown in Table I. Moreover, the predicted value at a certain point of time is derived from the observation values of the two or three preceding points. For instance, when the discrete step length $h$ is 20 minutes and the number of terms $n$ is 2, the predicted value at 15:20 for point 1 is derived from the observation values at 15:00 and 14:40. In addition, the extrapolated value of point 1 is derived from the observation values of the two or three preceding points. Hence, there are no corresponding extrapolated values for the first two or three observation points of the typhoon. The extrapolation situations for points 2 and 3 are similar.

The main contributions of this paper are listed as follows.

- We apply the Zhang extrapolation (ZE) formulas, based on Lagrange interpolation polynomials to predict typhoon

pathways using limited historical data, providing a practical approach for real-world scenarios with constrained data availability.

- We evaluate the effectiveness of the ZE formulas for predicting typhoon pathway points for three specific typhoons (typhoons Saudel, Molave, and Krovanh) using a dataset from the Wenzhou typhoon network (WTN).
- The study compares the performance of ZE formulas with other prediction methods, showing that the proposed formulas deliver higher precision, which enhances applications in oceanic safety and rescue operations.

## II. PROBLEM DESCRIPTION AND DATA SOURCE

This section introduces the background of the problem and the real data sources used to simulate the problem scenario.

### A. Problem Description

In real-world scenarios, various factors lead to limited data availability. When data is scarce, predicting the next data point in a time sequence becomes challenging. For instance, a ship encountering a typhoon at sea might not be able to receive external warning information and related prediction data due to severe weather conditions. Moreover, with a small amount of recorded typhoon path data, the ship needs to roughly estimate the next pathway point of the typhoon to avoid its impact as much as possible. In this paper, the dataset of typhoon paths near the coast of China is used to simulate the above scenario, applying the equally spaced time point sequence ZE formulas to predict future typhoon pathway points.

### B. Data Source

The data used in this paper comes from the dataset of typhoon paths near the coast of China collected by the Marine Science Data Center of the Chinese Academy of Sciences. This dataset is initially obtained from observational data on the Wenzhou typhoon network (http://www.wztf121.com/), which includes data on the latitude, longitude, intensity, pressure, central wind speed, movement speed, movement direction, Chinese names, English names, and numbers of typhoons at specific times. This paper uses the latitude and longitude data of typhoons, which are recorded at equal time intervals, meeting the equidistant extrapolation requirements of the equally spaced time point sequence ZE formulas.

### III. APPLICATION OF TYPHOON PATHWAY POINT PREDICTION

This section provides the pathway point prediction analyses for three typhoon instances. Each instance corresponds to the pathway trajectory of a typhoon, including horizontal comparisons of the extrapolation results for points 1, 2, and 3, as well as vertical comparisons for terms $n$ of 2 and 3. Additionally, the effect of predictions using ZE formulas on the entire dataset is presented and analyzed.

### A. Typhoon Case 1 (typhoon Saude)

In this subsection, the distance errors between the extrapolated points and the observed points are computed using great-circle distance. Using the average earth radius (approximately 6371.009 km) defined by the International Union of Geodesy and Geophysics, the method of great-circle distance computation is applied to compute the distance error between the extrapolated point and the observed point. The use of ellipsoid model can improve the calculation accuracy of the distance error, but in this application, it is more inclined to determine whether the error is within the range of the typhoon, so the great circle distance, which is simpler to calculate, is chosen. Additionally, for evaluating the extrapolation results, the average distance error (ADE) is used, defined as follows.

Definition 1: For $m$ data points, the ADE is defined as

$$\text{ADE} = \frac{1}{m} \sum_{i=1}^{m} e_i, \tag{1}$$

where $e_i$ is the distance error between the extrapolated point and the observed point at point of time $i$.

In this case, the data of typhoon Saudel is used, with 127 observed data points. Firstly, we analyze Zhang extrapolation for point 1, with the number of terms $n$ taken as 2 and 3. Part of results are shown in Table II and Table III, respectively. From the perspective of the extrapolation results at point 1 with 2 terms of longitude, the absolute error (AE) is mostly below 0.5, with an average AE being 0.2. From the perspective of the extrapolation results at point 1 with 2 terms of latitude, the AE is mostly below 0.3, with an average AE being 0.13. A comprehensive error comparison, as shown in Table II and Table III, indicates that the distance errors between the extrapolated points and the observed points are within the acceptable range. For instance, the ADE of the extrapolated point 1 with 2 terms is 14.49 nautical miles. When all data points (including data not shown in Table II) are considered, the distance error is 5.65388 nautical miles. The ADE of the extrapolated point 1 with 3 terms is 23.32 nautical miles. When all data points (including data not shown in Table III) are considered, the distance error is 9.191165 nautical miles. Although the ADE of 3 terms has risen compared with 2 terms, it is still within an acceptable range when the data is sufficient, and the number of terms can be decided upon revalidation after the initial results.

As shown in Table IV, the ADE of the extrapolated point 2 is 22.94 nautical miles, and 10.13 nautical miles when computed from all data points (including data not shown in Table IV). The ADE of the extrapolated point 3 is 32.16 nautical miles, and 14.25 nautical miles when computed from all data points (including data not shown in Table V). The distance errors of points 2 and 3 are still within an acceptable range, sufficient to avoid the typhoon path. For points 1, 2, and 3, the ADE of point 1 is the smallest, while that of point 3 is the largest. In practice, if data is sufficient, point 1, point 2, and point 3 are all extrapolated once. After integrating the results, the final judgment is made.

TABLE II
ZE Analysis at Point 1 (with $n = 2$) of Partial Data Points for Typhoon Saudel

| Time | Longitude | Latitude | Longitude (Extrapolated) | Latitude (Extrapolated) | AE (Longitude) | AE (Latitude) | Distance Error |
|---|---|---|---|---|---|---|---|
| 2020-10-19 14:00:00 | 128.1 | 13.9 | 127.8 | 14.0 | 0.3 | 0.2 | 18.48 |
| 2020-10-19 17:00:00 | 127.7 | 14.1 | 127.6 | 14.2 | 0.1 | 0.1 | 8.36 |
| 2020-10-19 20:00:00 | 127.2 | 14.5 | 127.3 | 14.3 | 0.3 | 0.2 | 21.18 |
| 2020-10-19 23:00:00 | 126.4 | 14.7 | 126.3 | 14.9 | 0.1 | 0.2 | 13.34 |
| 2020-10-20 02:00:00 | 125.8 | 14.8 | 125.8 | 14.9 | 0.0 | 0.1 | 8.35 |
| 2020-10-20 05:00:00 | 125.4 | 15.1 | 125.4 | 14.9 | 0.0 | 0.2 | 16.70 |
| 2020-10-20 08:00:00 | 124.6 | 15.4 | 124.5 | 15.4 | 0.1 | 0.0 | 5.79 |
| 2020-10-20 11:00:00 | 124.0 | 15.6 | 123.9 | 15.7 | 0.1 | 0.1 | 8.34 |
| 2020-10-20 14:00:00 | 123.5 | 15.9 | 123.2 | 15.8 | 0.3 | 0.1 | 18.34 |
| 2020-10-20 17:00:00 | 122.7 | 16.3 | 123.1 | 16.2 | 0.4 | 0.2 | 26.01 |

[1] The unit for distance error is the nautical mile.
[2] The distance error is computed as great-circle distance, with earth radius taken as 6371.009 km.

TABLE III
ZE Analysis at Point 1 (with $n = 3$) of Partial Data Points for Typhoon Saudel

| Time | Longitude | Latitude | Longitude (Extrapolated) | Latitude (Extrapolated) | AE (Longitude) | AE (Latitude) | Distance Error |
|---|---|---|---|---|---|---|---|
| 2020-10-19 17:00:00 | 127.7 | 14.1 | 127.9 | 14.1 | 0.2 | 0.0 | 11.65 |
| 2020-10-19 20:00:00 | 127.4 | 14.5 | 127.4 | 14.2 | 0.0 | 0.3 | 29.42 |
| 2020-10-19 23:00:00 | 126.7 | 14.7 | 126.2 | 14.6 | 0.5 | 0.1 | 33.40 |
| 2020-10-20 02:00:00 | 125.9 | 14.8 | 125.9 | 14.7 | 0.0 | 0.1 | 6.01 |
| 2020-10-20 05:00:00 | 125.4 | 15.1 | 125.3 | 15.0 | 0.1 | 0.1 | 25.05 |
| 2020-10-20 08:00:00 | 124.6 | 15.4 | 124.3 | 15.4 | 0.3 | 0.0 | 21.11 |
| 2020-10-20 11:00:00 | 123.9 | 15.6 | 123.7 | 15.6 | 0.2 | 0.0 | 20.33 |
| 2020-10-20 14:00:00 | 123.5 | 15.9 | 123.5 | 15.9 | 0.0 | 0.0 | 3.03 |
| 2020-10-20 17:00:00 | 122.7 | 16.3 | 123.1 | 16.0 | 0.4 | 0.3 | 44.21 |

TABLE IV
ZE Analysis at Point 2 (with $n = 2$) of Partial Data Points for Typhoon Saudel

| Time | Longitude | Latitude | Longitude (Extrapolated) | Latitude (Extrapolated) | AE (Longitude) | AE (Latitude) | Distance Error |
|---|---|---|---|---|---|---|---|
| 2020-10-19 17:00:00 | 127.7 | 14.1 | 127.4 | 14.0 | 0.3 | 0.1 | 44.54 |
| 2020-10-19 20:00:00 | 127.4 | 14.5 | 127.1 | 14.5 | 0.3 | 0.0 | 5.81 |
| 2020-10-19 23:00:00 | 126.4 | 14.7 | 126.9 | 14.6 | 0.5 | 0.1 | 31.43 |
| 2020-10-20 02:00:00 | 125.9 | 14.8 | 125.9 | 14.7 | 0.0 | 0.1 | 0.00 |
| 2020-10-20 05:00:00 | 125.4 | 15.1 | 125.2 | 15.1 | 0.2 | 0.0 | 8.34 |
| 2020-10-20 08:00:00 | 124.6 | 15.4 | 124.5 | 15.4 | 0.1 | 0.0 | 29.65 |
| 2020-10-20 11:00:00 | 123.9 | 15.6 | 123.8 | 15.7 | 0.1 | 0.1 | 8.34 |
| 2020-10-20 14:00:00 | 123.5 | 15.9 | 123.4 | 15.9 | 0.1 | 0.0 | 5.73 |
| 2020-10-20 17:00:00 | 122.7 | 16.3 | 123.1 | 16.0 | 0.4 | 0.3 | 11.54 |
| 2020-10-20 20:00:00 | 122.0 | 16.0 | 122.4 | 16.3 | 0.4 | 0.3 | 0.00 |
| 2020-10-20 23:00:00 | 121.9 | 16.0 | 122.7 | 16.5 | 0.8 | 0.5 | 55.02 |

TABLE V
ZE Analysis at Point 3 (with $n = 2$) of Partial Data Points for Typhoon Saudel

| Time | Longitude | Latitude | Longitude (Extrapolated) | Latitude (Extrapolated) | AE (Longitude) | AE (Latitude) | Distance Error |
|---|---|---|---|---|---|---|---|
| 2020-10-19 20:00:00 | 127.0 | 14.5 | 126.2 | 14.8 | 0.8 | 0.3 | 49.84 |
| 2020-10-19 23:00:00 | 126.4 | 14.7 | 126.6 | 14.8 | 0.2 | 0.1 | 5.74 |
| 2020-10-20 02:00:00 | 125.9 | 14.8 | 126.3 | 14.7 | 0.4 | 0.1 | 41.77 |
| 2020-10-20 05:00:00 | 125.4 | 15.1 | 124.5 | 15.3 | 0.9 | 0.2 | 35.35 |
| 2020-10-20 08:00:00 | 124.6 | 15.4 | 124.6 | 15.3 | 0.0 | 0.1 | 6.00 |
| 2020-10-20 11:00:00 | 123.9 | 15.6 | 124.4 | 15.5 | 0.5 | 0.1 | 39.99 |
| 2020-10-20 14:00:00 | 123.5 | 15.9 | 123.4 | 16.0 | 0.1 | 0.1 | 8.34 |
| 2020-10-20 17:00:00 | 122.7 | 16.3 | 123.1 | 16.0 | 0.4 | 0.3 | 11.54 |
| 2020-10-20 20:00:00 | 122.0 | 16.0 | 122.7 | 16.3 | 0.7 | 0.3 | 13.32 |
| 2020-10-20 23:00:00 | 121.6 | 16.0 | 122.3 | 16.8 | 0.7 | 0.8 | 79.51 |

TABLE VI
ZE ANALYSIS AT POINT 1 (WITH $n = 2$) OF PARTIAL DATA POINTS FOR TYPHOON MOLAVE

| Time | Longitude | Latitude | Longitude (Extrapolated) | Latitude (Extrapolated) | AE (Longitude) | AE (Latitude) | Distance Error |
|---|---|---|---|---|---|---|---|
| 2020-10-24 23:00:00 | 128.3 | 13.5 | 128.3 | 13.6 | 0.0 | 0.1 | 6.00 |
| 2020-10-25 02:00:00 | 127.8 | 13.4 | 127.6 | 13.5 | 0.2 | 0.1 | 16.75 |
| 2020-10-25 05:00:00 | 127.3 | 13.7 | 127.0 | 13.7 | 0.3 | 0.0 | 26.37 |
| 2020-10-25 08:00:00 | 126.3 | 13.6 | 125.9 | 14.0 | 0.4 | 0.4 | 24.71 |
| 2020-10-25 11:00:00 | 125.6 | 13.6 | 125.4 | 13.9 | 0.2 | 0.3 | 15.94 |
| 2020-10-25 14:00:00 | 124.8 | 13.4 | 124.4 | 13.7 | 0.4 | 0.3 | 5.84 |
| 2020-10-25 17:00:00 | 124.3 | 13.3 | 124.1 | 13.5 | 0.2 | 0.2 | 13.00 |
| 2020-10-25 20:00:00 | 123.8 | 13.4 | 123.4 | 13.6 | 0.4 | 0.2 | 13.32 |
| 2020-10-25 23:00:00 | 122.3 | 13.4 | 122.4 | 13.4 | 0.1 | 0.0 | 4.00 |
| 2020-10-26 02:00:00 | 121.8 | 13.2 | 121.3 | 13.2 | 0.5 | 0.0 | 40.92 |

TABLE VII
ZE ANALYSIS AT POINT 1 (WITH $n = 3$) OF PARTIAL DATA POINTS FOR TYPHOON MOLAVE

| Time | Longitude | Latitude | Longitude (Extrapolated) | Latitude (Extrapolated) | AE (Longitude) | AE (Latitude) | Distance Error |
|---|---|---|---|---|---|---|---|
| 2020-10-25 02:00:00 | 127.8 | 13.4 | 127.6 | 13.5 | 0.2 | 0.1 | 13.13 |
| 2020-10-25 05:00:00 | 127.3 | 13.7 | 127.5 | 13.5 | 0.2 | 0.2 | 46.37 |
| 2020-10-25 08:00:00 | 126.3 | 13.6 | 125.9 | 14.4 | 0.4 | 0.8 | 53.39 |
| 2020-10-25 11:00:00 | 125.6 | 13.6 | 125.1 | 14.0 | 0.5 | 0.4 | 24.72 |
| 2020-10-25 14:00:00 | 124.8 | 13.4 | 124.3 | 14.0 | 0.5 | 0.6 | 5.84 |
| 2020-10-25 17:00:00 | 124.3 | 13.3 | 123.8 | 14.1 | 0.5 | 0.8 | 18.33 |
| 2020-10-25 20:00:00 | 123.8 | 13.4 | 123.4 | 13.8 | 0.4 | 0.4 | 18.53 |
| 2020-10-25 23:00:00 | 122.3 | 13.4 | 122.8 | 14.1 | 0.5 | 0.7 | 83.8 |
| 2020-10-26 02:00:00 | 121.8 | 13.2 | 122.3 | 13.8 | 0.5 | 0.6 | 52.96 |

## B. Typhoon Case 2 (typhoon Molave)

The study of this case pertains to typhoon Molave, which includes a total of 66 observation points. Compared with typhoon Saudel, typhoon Molave has a shorter duration and fewer observation data points, making it a supplementary verification for case 1.

As shown in Table VI and TableVII, the ADEs between the point-1 extrapolated values and the observed values for the 2-term and 3-term extrapolations are 16.77 nautical miles and 30.61 nautical miles, respectively. Similar to case 1, the error of the 2-term point-1 extrapolation is lower than that of the 3-term extrapolation. When using all observation data for typhoon Molave (not just the data in the tables), the ADE for the 2-term extrapolation is 9.43 nautical miles, while for the 3-term extrapolation, it is 16.37 nautical miles. This is an increase in the point-1 extrapolation distance error compared with typhoon Saudel. Considering that each typhoon varies in intensity and direction and that the observation data points are relatively few, some fluctuations in the average extrapolation error are normal. Moreover, compared with the impact range of the typhoon, the point-1 extrapolation error in this case is still low, indicating that the point-1 ZE formula is also effective for typhoon Molave.

As shown in Table VIII, the distance error of the point-2 extrapolation fluctuates significantly, with a maximum of 70.42 nautical miles and a minimum of 5.83 nautical miles. For larger typhoons, 70.42 nautical miles is still within the impact range, making the point-2 extrapolation somewhat valuable for evasion. However, for smaller typhoons, 70.42 nautical miles exceeds the impact range, and its evasion value may be limited. The ADE in Table VIII is 28.38 nautical miles,

and it is 16.52 nautical miles when computed from all data points (including data not shown in Table VIII). In summary, from the perspective of ADE, the point-2 extrapolation still has some value for typhoon evasion. Due to the large errors of some data points, it is deduced that the point-2 extrapolation might be more effective for larger typhoons.

For typhoon Molave, as shown in Table IX, the ADE of the point-3 extrapolation is 36.65 nautical miles, and it is 22.58 nautical miles when computed from all data points (including data not shown in Table IX). Therefore, from the overall data performance, the point-3 extrapolation is meaningful for predicting typhoon pathway points. However, for individual points, such as the point with the largest error, the ADE reaches 100.14 nautical miles, indicating that the reference value of this extrapolation point is low. In brief, the point-3 extrapolation is more meaningful for the overall estimation of the typhoon pathway. For predicting individual pathway points, the accuracy may fluctuate significantly, and if conditions permit, the point-1 extrapolation should be chosen.

## C. Typhoon Case 3 (typhoon Krovanh)

The study of this case pertains to typhoon Krovanh, which includes a total of 34 observation data points. As shown in Table X and Table XI, the ADEs of the point-1 extrapolated values with 2 terms and 3 terms are 27.55 nautical miles and 36.63 nautical miles, respectively. When using all observation data of typhoon Krovanh (not just the data in the tables), the ADE for the 2-term extrapolation is 13.40 nautical miles, while for the 3-term extrapolation, it is 20.49 nautical miles. From the perspective of average data, the error of the 2-term point-1 extrapolation is lower than that of the 3-term extrapolation. As shown in Table XII, the distance error of the

| Time | Longitude | Latitude | Longitude (Extrapolated) | Latitude (Extrapolated) | AE (Longitude) | AE (Latitude) | Distance Error |
|---|---|---|---|---|---|---|---|
| 2020-10-25 02:00:00 | 127.8 | 13.4 | 127.6 | 13.8 | 0.2 | 0.4 | 26.70 |
| 2020-10-25 05:00:00 | 127.3 | 13.7 | 126.9 | 13.7 | 0.4 | 0.0 | 5.83 |
| 2020-10-25 08:00:00 | 126.3 | 13.6 | 126.0 | 13.5 | 0.3 | 0.1 | 37.81 |
| 2020-10-25 11:00:00 | 125.6 | 13.5 | 125.4 | 13.8 | 0.2 | 0.3 | 49.43 |
| 2020-10-25 14:00:00 | 124.8 | 13.4 | 124.2 | 13.3 | 0.6 | 0.1 | 5.84 |
| 2020-10-25 17:00:00 | 124.3 | 13.3 | 124.0 | 13.4 | 0.3 | 0.1 | 5.00 |
| 2020-10-25 20:00:00 | 123.8 | 13.4 | 123.3 | 13.2 | 0.5 | 0.2 | 21.25 |
| 2020-10-25 23:00:00 | 122.3 | 13.4 | 123.4 | 13.4 | 1.1 | 0.0 | 21.25 |
| 2020-10-26 02:00:00 | 121.8 | 13.2 | 121.3 | 13.1 | 0.5 | 0.1 | 70.42 |

| Time | Longitude | Latitude | Longitude (Extrapolated) | Latitude (Extrapolated) | AE (Longitude) | AE (Latitude) | Distance Error |
|---|---|---|---|---|---|---|---|
| 2020-10-25 05:00:00 | 127.0 | 13.7 | 126.9 | 14.0 | 0.1 | 0.3 | 18.93 |
| 2020-10-25 08:00:00 | 126.3 | 13.6 | 126.2 | 13.8 | 0.1 | 0.2 | 13.35 |
| 2020-10-25 11:00:00 | 125.6 | 13.5 | 126.2 | 14.4 | 0.6 | 0.9 | 47.43 |
| 2020-10-25 14:00:00 | 124.8 | 13.4 | 124.6 | 14.6 | 0.2 | 1.2 | 72.98 |
| 2020-10-25 17:00:00 | 124.3 | 13.3 | 124.2 | 13.3 | 0.1 | 0.0 | 5.00 |
| 2020-10-25 20:00:00 | 123.8 | 13.4 | 123.5 | 13.5 | 0.3 | 0.1 | 16.76 |
| 2020-10-25 23:00:00 | 122.3 | 13.4 | 123.5 | 14.1 | 1.2 | 0.7 | 16.76 |
| 2020-10-26 02:00:00 | 121.8 | 13.2 | 122.8 | 13.7 | 1.0 | 0.5 | 21.02 |
| 2020-10-26 08:00:00 | 121.2 | 13.0 | 121.3 | 12.9 | 0.1 | 0.1 | 100.14 |

point-2 extrapolation fluctuates significantly, with a maximum of 151.74 nautical miles and a minimum of 6.00 nautical miles. The 151.74 nautical miles exceeds the impact range of most typhoons, making its evasion value limited. However, from the perspective of overall data, most of the distance errors are within the typhoon's impact range. The ADE in Table XII is 47.54 nautical miles, and it is 24.76 nautical miles when computed from all data points (including data not shown in Table XII). In summary, from the perspective of ADE, the point-2 extrapolation still has some value for typhoon evasion. Due to the large errors for some data points, it can be concluded that the point-2 extrapolation might be more effective for larger typhoons.

As shown in Table XIII , the ADE for the point-3 extrapolation is 65.57 nautical miles, and it is 36.11 nautical miles when computed from all data points (including data not shown in Table XIII). Therefore, from the overall data performance, the point-3 extrapolation is meaningful for predicting typhoon pathway points.

### D. Statistical Results

In this subsection, all available data in the extrapolated dataset are statistically analyzed to evaluate the effectiveness and applicability of ZE formulas for points 1, 2, and 3. The typhoon observation data from the dataset are recorded at equal time intervals, with records at 6-hour, 3-hour, and 1-hour intervals. Specifically, there are 1451 typhoons recorded at 6-hour intervals, 189 at 3-hour intervals, and 105 at 1-hour intervals.

*1) Point-1 Extrapolation Data Statistics:* For typhoon records with 6-hour intervals, the 2-term point-1 extrapolation is performed, resulting in 44,116 extrapolated point values,

and their corresponding distance errors are computed. The ADE is 23.39 nautical miles, and the overall statistical results are shown in Figs. 1 and 2. If an error within 60 nautical miles is considered as an effective prediction, the efficiency rate is 95%. Moreover, Fig. 1 reveals that 24,928 instances (57%) have a distance error of less than 20 nautical miles. It indicates that with only two observation points, the point-1 extrapolation has about a 57% chance of yielding a highly accurate pathway point prediction. There are 12,753 instances (29%) with a distance error between 20 and 40 nautical miles, showing that the point-1 extrapolation has about a 29% chance of yielding a reasonably good pathway point prediction. The detailed distribution data can be seen in Figs. 1 and 2.

For large-scale natural disasters like typhoons, a prediction error exceeding 100 nautical miles provides little help for evasion. However, the proportion of errors exceeding 100 nautical miles is only about 1%. Moreover, the proportion of distance errors below 40 nautical miles is 86%. That means under extreme conditions (without external help and with limited recorded data), the point-1 ZE formula still has over an 80% chance of achieving a high-level prediction. In summary, when the discrete interval is 6 hours, the point-1 ZE formula is effective for predicting typhoon pathway points.

The point-1 extrapolation results with discrete interval $\hat{h}$ being 3 hours and 1 hour are shown in Table XIV. The ADEs for the point-1 extrapolation with time intervals of 3 hours and 1 hour are 18.10 nautical miles and 6.95 nautical miles respectively, both of which have decreased compared with the 6-hour interval. When the time interval is 3 hours, the proportion of distance errors below 40 nautical miles is 92.42%. While for 1-hour interval, it reaches 99.71%. This illustrates the effectiveness of the point-1 ZE formula and

TABLE X
ZE ANALYSIS AT POINT 1 (WITH $n = 2$) OF PARTIAL DATA POINTS FOR TYPHOON KROVANH

| Time | Longitude | Latitude | Longitude (Extrapolated) | Latitude (Extrapolated) | AE (Longitude) | AE (Latitude) | Distance Error |
|---|---|---|---|---|---|---|---|
| 2020-12-19 05:00:00 | 121.3 | 9.2 | 120.2 | 10.1 | 1.10 | 0.90 | 84.61 |
| 2020-12-19 08:00:00 | 121.9 | 9.3 | 121.3 | 9.2 | 0.30 | 0.10 | 18.76 |
| 2020-12-19 11:00:00 | 120.7 | 9.4 | 120.4 | 9.5 | 0.30 | 0.10 | 13.28 |
| 2020-12-19 14:00:00 | 119.7 | 9.5 | 119.4 | 9.4 | 0.30 | 0.10 | 13.28 |
| 2020-12-19 17:00:00 | 119.8 | 9.7 | 119.9 | 9.5 | 0.10 | 0.20 | 5.87 |
| 2020-12-19 20:00:00 | 118.9 | 9.6 | 119.2 | 9.5 | 0.30 | 0.10 | 18.73 |
| 2020-12-19 23:00:00 | 117.9 | 10.4 | 118.7 | 9.9 | 0.80 | 0.50 | 59.37 |
| 2020-12-20 02:00:00 | 116.7 | 10.7 | 117.4 | 10.4 | 0.70 | 0.30 | 44.58 |
| 2020-12-20 05:00:00 | 115.2 | 11.1 | 115.8 | 11.0 | 0.60 | 0.10 | 42.00 |
| 2020-12-20 08:00:00 | 114.8 | 10.9 | 114.7 | 10.8 | 0.10 | 0.10 | 8.34 |
| 2020-12-20 11:00:00 | 114.0 | 10.9 | 113.7 | 10.9 | 0.30 | 0.00 | 18.70 |
| 2020-12-20 14:00:00 | 113.2 | 10.9 | 113.4 | 10.8 | 0.20 | 0.10 | 26.76 |

TABLE XI
ZE ANALYSIS AT POINT 1 (WITH $n = 3$) OF PARTIAL DATA POINTS FOR TYPHOON KROVANH

| Time | Longitude | Latitude | Longitude (Extrapolated) | Latitude (Extrapolated) | AE (Longitude) | AE (Latitude) | Distance Error |
|---|---|---|---|---|---|---|---|
| 2020-12-19 08:00:00 | 121.9 | 9.3 | 122.4 | 8.3 | 1.40 | 1.00 | 102.49 |
| 2020-12-19 11:00:00 | 120.7 | 9.4 | 120.4 | 9.5 | 0.30 | 0.10 | 18.75 |
| 2020-12-19 14:00:00 | 119.7 | 9.5 | 119.5 | 9.3 | 0.20 | 0.20 | 13.28 |
| 2020-12-19 17:00:00 | 119.8 | 9.7 | 120.0 | 9.6 | 0.20 | 0.10 | 21.56 |
| 2020-12-19 20:00:00 | 118.9 | 9.6 | 119.7 | 10.0 | 0.80 | 0.40 | 18.70 |
| 2020-12-19 23:00:00 | 117.9 | 10.4 | 117.7 | 10.5 | 0.20 | 0.10 | 8.34 |
| 2020-12-20 02:00:00 | 116.7 | 10.7 | 117.3 | 11.0 | 0.60 | 0.30 | 41.70 |
| 2020-12-20 05:00:00 | 115.7 | 10.9 | 116.4 | 11.0 | 0.70 | 0.10 | 51.88 |
| 2020-12-20 08:00:00 | 114.8 | 11.0 | 115.9 | 10.9 | 1.10 | 0.10 | 59.37 |
| 2020-12-20 11:00:00 | 114.0 | 10.9 | 114.8 | 10.9 | 0.80 | 0.00 | 50.66 |
| 2020-12-20 14:00:00 | 113.2 | 10.9 | 113.7 | 10.9 | 0.50 | 0.00 | 42.00 |
| 2020-12-20 17:00:00 | 112.0 | 11.0 | 112.5 | 11.0 | 0.50 | 0.00 | 18.70 |
| 2020-12-20 20:00:00 | 111.0 | 11.0 | 111.5 | 11.0 | 0.50 | 0.00 | 12.01 |

TABLE XII
ZE ANALYSIS AT POINT 2 (WITH $n = 2$) OF PARTIAL DATA POINTS FOR TYPHOON KROVANH

| Time | Longitude | Latitude | Longitude (Extrapolated) | Latitude (Extrapolated) | AE (Longitude) | AE (Latitude) | Distance Error |
|---|---|---|---|---|---|---|---|
| 2020-12-19 08:00:00 | 121.9 | 9.3 | 119.1 | 11.1 | 1.90 | 1.70 | 151.74 |
| 2020-12-19 11:00:00 | 120.7 | 9.4 | 121.3 | 9.2 | 0.60 | 0.20 | 37.52 |
| 2020-12-19 14:00:00 | 119.7 | 9.5 | 120.4 | 9.6 | 0.70 | 0.10 | 23.68 |
| 2020-12-19 17:00:00 | 119.8 | 9.6 | 120.0 | 9.6 | 0.20 | 0.00 | 23.68 |
| 2020-12-19 20:00:00 | 118.9 | 9.6 | 119.7 | 10.0 | 0.80 | 0.40 | 34.88 |
| 2020-12-19 23:00:00 | 117.9 | 10.4 | 117.8 | 10.0 | 0.10 | 0.40 | 59.73 |
| 2020-12-20 02:00:00 | 116.7 | 10.7 | 117.7 | 10.6 | 1.00 | 0.10 | 37.42 |
| 2020-12-20 05:00:00 | 115.7 | 10.9 | 116.8 | 10.4 | 1.10 | 0.50 | 60.42 |
| 2020-12-20 08:00:00 | 114.8 | 11.0 | 117.7 | 11.8 | 2.90 | 0.80 | 174.73 |
| 2020-12-20 11:00:00 | 114.0 | 10.9 | 114.7 | 10.4 | 0.70 | 0.50 | 55.31 |
| 2020-12-20 14:00:00 | 113.2 | 10.9 | 113.4 | 11.0 | 0.20 | 0.10 | 8.34 |
| 2020-12-20 17:00:00 | 112.0 | 11.0 | 112.7 | 11.0 | 0.70 | 0.00 | 18.70 |
| 2020-12-20 20:00:00 | 111.0 | 11.0 | 111.5 | 11.0 | 0.50 | 0.00 | 6.00 |

TABLE XIII
ZE ANALYSIS AT POINT 3 (WITH $n = 2$) OF PARTIAL DATA POINTS FOR TYPHOON KROVANH

| Time | Longitude | Latitude | Longitude (Extrapolated) | Latitude (Extrapolated) | AE (Longitude) | AE (Latitude) | Distance Error |
|---|---|---|---|---|---|---|---|
| 2020-12-19 08:00:00 | 120.7 | 9.4 | 118.0 | 11.9 | 2.70 | 2.50 | 218.88 |
| 2020-12-19 11:00:00 | 120.7 | 9.4 | 121.3 | 9.2 | 0.60 | 0.20 | 66.27 |
| 2020-12-19 14:00:00 | 119.7 | 9.5 | 120.4 | 9.6 | 0.70 | 0.10 | 23.68 |
| 2020-12-19 17:00:00 | 119.8 | 9.6 | 120.0 | 9.6 | 0.20 | 0.00 | 23.68 |
| 2020-12-19 20:00:00 | 118.9 | 9.6 | 119.7 | 10.0 | 0.80 | 0.40 | 34.88 |
| 2020-12-19 23:00:00 | 117.9 | 10.4 | 117.8 | 10.0 | 0.10 | 0.40 | 59.73 |
| 2020-12-20 02:00:00 | 116.7 | 10.7 | 117.7 | 10.6 | 1.00 | 0.10 | 37.42 |
| 2020-12-20 05:00:00 | 115.7 | 10.9 | 116.8 | 10.4 | 1.10 | 0.50 | 60.42 |
| 2020-12-20 08:00:00 | 114.8 | 11.0 | 117.7 | 11.8 | 2.90 | 0.80 | 174.73 |
| 2020-12-20 11:00:00 | 114.0 | 10.9 | 114.7 | 10.4 | 0.70 | 0.50 | 55.31 |
| 2020-12-20 14:00:00 | 113.2 | 10.9 | 113.4 | 11.0 | 0.20 | 0.10 | 8.34 |
| 2020-12-20 17:00:00 | 112.0 | 11.0 | 112.7 | 11.0 | 0.70 | 0.00 | 18.70 |
| 2020-12-20 20:00:00 | 111.0 | 11.0 | 111.5 | 11.0 | 0.50 | 0.00 | 6.00 |

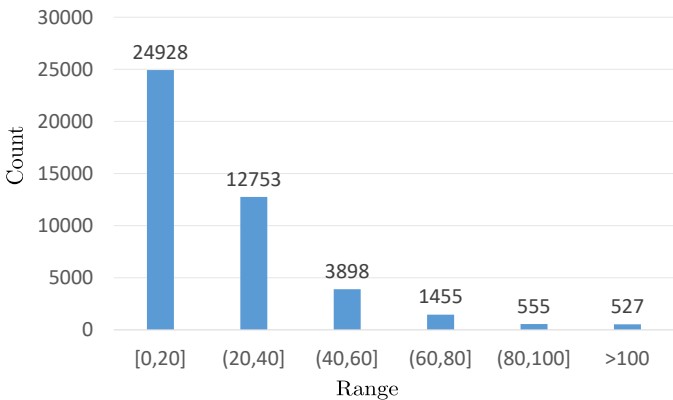

Fig. 1. Range Numbers of occurrences of point-1 ZE distance errors (with $\hat{h}$ = 6 hours and unit being nautical miles)

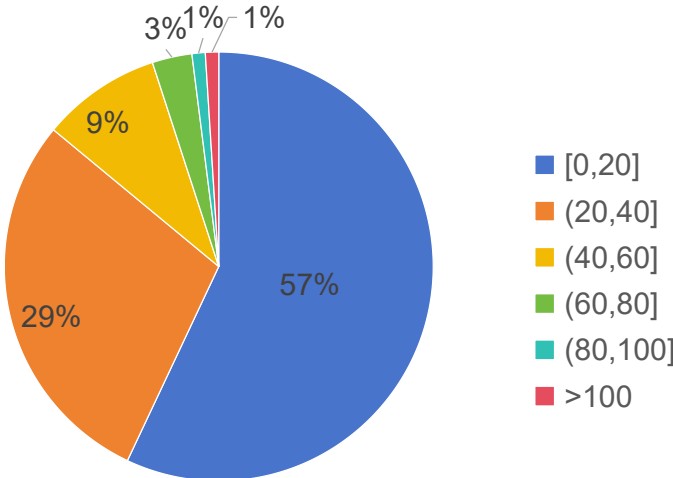

Fig. 2. Proportion of point-1 ZE distance errors (with $\hat{h}$ = 6 hours and unit being nautical miles)

indicates that the smaller the time interval is, the more effective the point-1 extrapolation results are.

*2) Point-2 Extrapolation Data Statistics:* The point-2 extrapolation results are shown in Table XV. From the overall data performance, the ADE for the point-2 extrapolation with a 6-hour time interval is 50.82 nautical miles, while 32.29 nautical miles for a 3-hour interval and 13.10 nautical miles for a 1-hour interval. The ADE is relatively high when the time interval is 6 hours. But considering the scale of

TABLE XIV
STATISTICAL ANALYSIS OF POINT-1 ZHANG EXTRAPOLATION BASED ON 2 TERMS (WITH INTERVALS BEING 3 HOURS AND 1 HOUR)

| 3 Hours (Total: 4865) | | | 1 Hour (Total: 10871) | | |
|---|---|---|---|---|---|
| Error Interval | Count | Frequency (%) | Error Interval | Count | Frequency (%) |
| [0, 20] | 3291 | 67.65 | [0, 20] | 10585 | 97.37 |
| (20, 40] | 1205 | 24.77 | (20, 40] | 254 | 2.34 |
| (40, 60] | 269 | 5.53 | (40, 60] | 19 | 0.17 |
| (60, 80] | 65 | 1.34 | (60, 80] | 7 | 0.06 |
| (80, 100] | 26 | 0.53 | (80, 100] | 2 | 0.02 |
| (100, +∞) | 9 | 0.18 | (100, +∞) | 4 | 0.04 |

typhoons, it still holds some significance for avoiding the typhoon. Therefore, the point-2 ZE formula is effective for predicting typhoon pathway points. With a 6-hour interval, 70.72% of the distance errors are within 60 nautical miles. Although the prediction accuracy decreases when the distance error exceeds 60 nautical miles, the results still hold some significance considering the scale of typhoons. When using a 3-hour interval, 88.66% of the predictions have distance errors within 60 nautical miles. This is an increase compared with the 6-hour interval, indicating improved accuracy of the point-2 extrapolation. When using a 1-hour interval, 83.66% of the distance errors are within 20 nautical miles, and 97.47% are within 60 nautical miles. Although the accuracy decreases compared with the point-1 extrapolation results, it is still sufficient for predicting typhoon pathway points. In summary, the point-2 ZE formula is effective.

*3) Point-3 Extrapolation Data Statistics:* The point-3 extrapolations are performed using time intervals of 6 hours, 3 hours, and 1 hour on the dataset, and the results are shown in Table XVI. The ADE for the point-3 extrapolation with a 6-hour time interval is 86.84 nautical miles, while 48.86 nautical miles for a 3-hour interval and 19.66 nautical miles for a 1-hour interval. When using a 6-hour interval, the ADE reaches 86.84 nautical miles. Although this still holds some significance, a smaller time interval should be chosen when conditions permit. With a 3-hour interval, 71.97% of the distance errors are within 60 nautical miles, indicating improved accuracy compared with the 6-hour interval. When using a 1-hour interval, 83.66% of the distance errors are within 20 nautical miles, and 97.47% are within 60 nautical miles. Therefore, when performing point-3 extrapolations, data should be recorded at the smallest possible time intervals. In summary, under suitable conditions, the point-3 ZE formula is effective for predicting typhoon pathway points.

## IV. COMPARISON WITH OTHER EXTRAPOLATION FORMULA

This section provides a comparison with previous similar work. [7] presents an extrapolation formula that can be used for predicting typhoon pathway points, but the derivation involves the acceleration of the typhoon path center, making the formula different from the one proposed in this paper. To facilitate comparison, we first convert its form to be consistent with ZE formula. The 12-hour ahead typhoon path prediction formula given in [7] is

$$\phi_{12} = \phi_0 + 4\Delta\phi_0 - \Delta\phi_{-12}, \qquad (2)$$

where $\phi_{12}$ represents the 12-hour ahead extrapolation result, $\phi_0$ represents the current observation data, $\phi_{-12}$ represents the observation data 12 hours ago, $\Delta\phi_0$ represents the current acceleration, and $\Delta\phi_{-12}$ represents the acceleration 12 hours ago. With a time interval of 6 hours, the 12-hour ahead path prediction corresponds to the point-2 extrapolation. Combining the definition of acceleration in [7], formula (2) can be simplified and rearranged to be consistent with the form of the formula proposed in this paper:

TABLE XV
STATISTICAL ANALYSIS OF POINT-2 ZE FORMULA BASED ON 2 ITEMS (WITH INTERVALS OF 6 HOURS, 3 HOURS, AND 1 HOUR)

| 6 Hours (Total: 42526) | | | 3 Hours (Total: 3891) | | | 1 Hour (Total: 10623) | | |
|---|---|---|---|---|---|---|---|---|
| **Error Interval** | **Count** | **Frequency (%)** | **Error Interval** | **Count** | **Frequency (%)** | **Error Interval** | **Count** | **Frequency (%)** |
| [0, 20] | 8985 | 21.13 | [0, 20] | 1385 | 35.59 | [0, 20] | 8887 | 83.66 |
| (20, 40] | 12519 | 29.44 | (20, 40] | 1461 | 37.55 | (20, 40] | 1524 | 12.35 |
| (40, 60] | 8571 | 20.15 | (40, 60] | 601 | 15.45 | (40, 60] | 155 | 1.46 |
| (60, 80] | 5128 | 12.06 | (60, 80] | 262 | 6.73 | (60, 80] | 39 | 0.37 |
| (80, 100] | 2878 | 6.77 | (80, 100] | 93 | 2.39 | (80, 100] | 9 | 0.08 |
| (100, +∞) | 4445 | 10.45 | (100, +∞) | 86 | 2.21 | (100, +∞) | 9 | 0.09 |

\* $\hat{h}$ represents time interval.

TABLE XVI
STATISTICAL ANALYSIS OF POINT-3 ZE FORMULA BASED ON 2 ITEMS (WITH INTERVALS OF 6 HOURS, 3 HOURS, AND 1 HOUR)

| 6 Hours (Total: 40966) | | | 3 Hours (Total: 3564) | | | 1 Hour (Total: 10377) | | |
|---|---|---|---|---|---|---|---|---|
| **Error Interval** | **Count** | **Frequency (%)** | **Error Interval** | **Count** | **Frequency (%)** | **Error Interval** | **Count** | **Frequency (%)** |
| [0, 20] | 2527 | 6.17 | [0, 20] | 638 | 17.90 | [0, 20] | 6406 | 61.73 |
| (20, 40] | 5903 | 14.41 | (20, 40] | 1140 | 31.97 | (20, 40] | 2481 | 23.91 |
| (40, 60] | 7381 | 18.02 | (40, 60] | 710 | 19.93 | (40, 60] | 1260 | 12.14 |
| (60, 80] | 10820 | 26.40 | (60, 80] | 237 | 6.65 | (60, 80] | 154 | 1.48 |
| (80, 100] | 13240 | 32.30 | (80, 100] | 567 | 15.90 | (80, 100] | 34 | 0.33 |
| (100, +∞) | 12600 | 30.76 | (100, +∞) | 272 | 7.63 | (100, +∞) | 34 | 0.33 |

$$\theta_{k+2} = 4\theta_k - 4\theta_{k-1} + \theta_{k-2}. \tag{3}$$

When the time interval is 3 hours (or 1 hour), formula (3) can predict the typhoon pathway points for the next 6 hours (or 2 hours). Based on formula (3), typhoon pathway points are predicted in the dataset for comparison with the work in this paper. According to the form of formula (3), it is clear that it corresponds to the point-2 ZE formula. When the number of terms for the point-2 ZE formula is chosen as 2, it becomes:

$$\theta_{k+2} = 3\theta_k - 2\theta_{k-1}. \tag{4}$$

Tables XVII, XVIII, and XIX correspond to the results of extrapolation comparison with time intervals being 6 hours, 3 hours, and 1 hour, respectively. In the extrapolation comparison with a 6-hour time interval, the ADE of formula (3) (corresponding to the extrapolation formula from [7]) is 57.12 nautical miles, while the ADE for the point-2 ZE formula is 50.82 nautical miles. Therefore, from the perspective of ADE, the point-2 ZE formula has higher accuracy. As shown in Table XIX, in the intervals [0, 20] and (20, 40], the frequency of the point-2 ZE formula is higher than that of formula (3). This indicates that the results of the point-2 ZE formula are more concentrated in high-accuracy and relatively high-accuracy intervals, providing more useful predictive information for emergency avoidance.

In the extrapolation comparison with a 3-hour interval, the ADE of formula (3) is 42.54 nautical miles, while the ADE for the point-2 ZE formula is 32.29 nautical miles, indicating that the point-2 ZE formula has a lower extrapolation error. As shown in Table XVIII, in the [0, 20] interval, the frequency of the point-2 ZE formula is higher than that of formula (3).

TABLE XVII
COMPARISON OF EXTRAPOLATION FORMULA (3) WITH POINT-2 ZE FORMULA (WITH $h$ = 1 HOUR)

| Extrapolation Formula (3) | | | Point-2 ZE Formula | | |
|---|---|---|---|---|---|
| Error Interval | Count | Frequency (%) | Error Interval | Count | Frequency (%) |
| [0, 20] | 7052 | 67.96 | [0, 20] | 8887 | 83.66 |
| (20, 40] | 2783 | 26.82 | (20, 40] | 1524 | 12.35 |
| (40, 60] | 409 | 3.94 | (40, 60] | 155 | 1.46 |
| (60, 80] | 84 | 0.81 | (60, 80] | 39 | 0.37 |
| (80, 100] | 25 | 0.24 | (80, 100] | 8 | 0.08 |
| (100, +∞) | 27 | 0.26 | (100, +∞) | 10 | 0.09 |

TABLE XVIII
COMPARISON OF EXTRAPOLATION FORMULA (3) WITH POINT-2 ZE FORMULA (WITH $h$ = 3 HOURS)

| Extrapolation Formula (3) | | | Point-2 ZE Formula | | |
|---|---|---|---|---|---|
| Error Interval | Count | Frequency (%) | Error Interval | Count | Frequency (%) |
| [0, 20] | 887 | 24.89 | [0, 20] | 1385 | 35.59 |
| (20, 40] | 1186 | 33.28 | (20, 40] | 1461 | 37.55 |
| (40, 60] | 750 | 21.41 | (40, 60] | 601 | 15.45 |
| (60, 80] | 355 | 9.96 | (60, 80] | 262 | 6.73 |
| (80, 100] | 180 | 5.05 | (80, 100] | 93 | 2.39 |
| (100, +∞) | 206 | 5.78 | (100, +∞) | 86 | 2.21 |

TABLE XIX
COMPARISON OF EXTRAPOLATION FORMULA (3) WITH POINT-2 ZE FORMULA (WITH $h$ = 6 HOURS)

| Extrapolation Formula (3) | | | Point-2 ZE Formula | | |
|---|---|---|---|---|---|
| Error Interval | Count | Frequency (%) | Error Interval | Count | Frequency (%) |
| [0, 20] | 6955 | 16.98 | [0, 20) | 8985 | 21.13 |
| (20, 40] | 11303 | 27.59 | (20, 40] | 12519 | 29.44 |
| (40, 60] | 8553 | 20.88 | (40, 60] | 8571 | 20.15 |
| (60, 80] | 5448 | 13.30 | (60, 80] | 5128 | 12.06 |
| (80, 100] | 3212 | 7.84 | (80, 100] | 2878 | 6.77 |
| (100, +∞) | 5495 | 13.41 | (100, +∞) | 4445 | 10.45 |

Additionally, in the [0, 40] interval, compared with 58.17% for formula (3), the frequency of the point-2 ZE formula is 73.14%. Similar to the 6-hour interval extrapolation case, the results of the point-2 ZE formula are more concentrated in the high-accuracy and relatively high-accuracy intervals.

Similarly, in the extrapolation comparison with a 1-hour interval, similar deduction can be drawn as above. The ADE of formula (3) is 17.81 nautical miles, while the ADE for the point-2 ZE formula is 13.10 nautical miles. The point-2 ZE formula's ADE is 4.71 nautical miles lower than that of formula (3). As shown in Table XVII, in the [0, 20] interval, the frequency of the point-2 ZE formula is higher than that of formula (3). Additionally, in the [0, 40] interval, the frequency of the point-2 ZE formula is 96.01%, compared with 94.78% from formula (3). Similar to the 3-hour interval extrapolation case, the results of the point-2 ZE formula are more concentrated in the high-accuracy and relatively high-accuracy intervals.

In summary, the point-2 ZE formula has higher prediction accuracy for typhoon pathway points based on limited historical data.

## V. PAPER SUMMARY AND CONCLUSION

This paper has applied equal-interval time point sequence ZE formulas to the prediction of typhoon pathway points, aiming to predict future typhoon pathway points under extreme conditions (without external assistance and with limited recorded data) to help ships navigate and avoid typhoons. Initially, three typhoon cases have been introduced in detail and corresponding extrapolation analyses have been performed. The results indicate that the point-1, point-2, and point-3 extrapolations are all effective. Then, from the perspective of overall statistical data, the effectiveness of the point-1, point-2, and point-3 ZE formulas has been analyzed separately, and a predictive performance comparison between the point-2 ZE formula and the extrapolation formula from [7] has been conducted. The results show that the point-1, point-2, and point-3 ZE formulas all have achieved good results in typhoon pathway point prediction, with the point-1 extrapolation having the highest prediction accuracy. Therefore, when conditions permit, the point-1 extrapolation results should be preferred, or the point-2 and point-3 extrapolation results should be combined to judge the overall trajectory of the typhoon. In the comparison with the extrapolation formula from [7], the point-2 ZE formula proposed in this paper has higher prediction accuracy. With time intervals being 6 hours, 3 hours, and 1 hour respectively, the ADEs of the point-2 ZE formula are 6.30 nautical miles, 10.25 nautical miles, and 4.71 nautical miles , which are lower than those of the extrapolation formula from [7]. Moreover, compared with the extrapolation formula from [7], the extrapolation results of the point-2 ZE formula have been more concentrated in the high-accuracy and relatively high-accuracy intervals.

## ACKNOWLEDGEMENT

This paper is supported by the National Natural Science Foundation of China (with number 62376290). Besides, the corresponding author is Yunong Zhang.

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
