# OpenReview forum: "Prediction of Typhoon Pathway Points Based on Zhang Extrapolation (ZE) Formula for Points 1 to 3"
_IEEE.org/ICIST/2024/Conference — IEEE ICIST 2024 Conference Submission_

### Official Review · Reviewer_XusU · 2024-08-21
**This article is quite fascinating and of high quality**

**Rating:** 7
**Confidence:** 3

**Review:**

The paper titled "Prediction of Typhoon Pathway Points Based on Zhang Extrapolation (ZE) Formula for Points 1 to 3"  demonstrates the effectiveness of Zhang extrapolation (ZE) formulas. Statistical analysis indicates that, in comparison to other methods, the proposed ZE formula demonstrates higher accuracy, significantly enhancing its application in maritime safety and emergency response. My specific feedback is as follows: 1) What is the Zhang extrapolation (ZE) formulas? More introductions of it  should be given. 2) Some formatting issues need to be addressed.

---

### Official Review · Reviewer_HuDP · 2024-08-22
**This article is very interesting and a good one**

**Rating:** 7
**Confidence:** 3

**Review:**

This study evaluates the effectiveness of Zhang extrapolation (ZE) formulas, on the basis of equally spaced time point sequence, in predicting typhoon pathways using limited historical data. The theory is correct and can be accepted after responding the following comments.
(1) In the introduction, it is not enough to state the current work. It should be expended and reconstructed.
(2) In the end of Section 1, the organization of this study is suggested to be summarized.
(3) There are many typos and grammar errors. The authors should have a native English speaker or software packages to perform the editing check.
(4) In Conclusion, it is necessary to point out the limitations of the control approach and increase future research directions.

---

### Official Review · Reviewer_Mqam · 2024-08-23
**This study evaluates the effectiveness of Zhang extrapolation (ZE) formulas, on the basis of equally spaced time point sequence, in predicting typhoon pathways using limited historical data. The ZE formulas for 1 through 3 points are tested on typhoons Saudel, Molave, and Krovanh, using a dataset from the Wenzhou typhoon network (WTN). Statistical analyses reveal that the point-1 extrapolation consistently offers superior accuracy, with mean absolute error and great-circle distance metrics within practical limitations. Comparisons with other methods show the proposed ZE formulas deliver higher precision, making them highly enhance applications in maritime safety and emergency response. Comments for this submission are given as follows.**

**Rating:** 7
**Confidence:** 3

**Review:**

(1)The grammar of this article is very good, the article is well worded, but individual areas need further work and the writer should double check the grammar of this article.
(2)The references in this paper are very appropriate, but are they too few in number and the authors are invited to add them at their discretion.
(3)The literature citations in this paper are very appropriate, but whether the year of some of the literature is too old, for example, literature [10], the authors are requested to modify them at their discretion.

---

### Comment · Reviewer_HuDP · 2024-08-21
**This article is very interesting and a good one**

This study evaluates the effectiveness of Zhang extrapolation (ZE) formulas, on the basis of equally spaced time point sequence, in predicting typhoon pathways using limited historical data. The theory is correct and can be accepted after responding the following comments.
(1) In the introduction, it is not enough to state the current work. It should be expended and reconstructed.
(2) In the end of Section 1, the organization of this study is suggested to be summarized.
(3) There are many typos and grammar errors. The authors should have a native English speaker or software packages to perform the editing check.
(4) In Conclusion, it is necessary to point out the limitations of the control approach and increase future research directions.

---

### Decision · Program_Chairs · 2024-09-06

Accept (Oral)